# Experimental Study on Submerged Horizontal Perforated Plates under Irregular Wave Conditions

**Yanna Zheng [1,2], Yifan Zhou [3], Ruijia Jin [4], Yingna Mu [1,2,*], Ming He [5] and Lingxiao Zhao [1]**

1   College of Marine and Civil Engineering, Dalian Ocean University, Dalian 116023, China;
    zhengyn@dlou.edu.cn (Y.Z.); lingxiaoz@outlook.com (L.Z.)
2   Key Laboratory of Environment Controlled Aquaculture, Dalian Ocean University, Ministry of Education,
    Dalian 116023, China
3   Lishui Municipal Water Conservancy Bureau, Lishui 323000, China; 18100173763@163.com
4   Tianjin Research Institute for Water Transport Engineering, M.O.T., Tianjin 300456, China;
    ruijia_jin@163.com
5   Tianjin Key Laboratory of Port and Ocean Engineering, Tianjin University, Tianjin 300350, China;
    inghe@tju.edu.cn
*   Correspondence: myn-myn@dlou.edu.cn; Tel.: +86-0411-84763479 or +86-186-0428-1847

**Abstract:** This study presents novel analytical solutions for analyzing wave dissipation effect and bottom flow field characteristics of permeable submerged horizontal plates through physical model trials. The experimental results show that a solid submerged horizontal plate effectively attenuates wave cycles, with a greater periodic attenuation effect at smaller submerged depths. However, this attenuation effect becomes reduced or less pronounced after a certain threshold. Selecting an optimal opening ratio becomes key to achieving the desired cycle attenuation. When the inundation depth of the horizontal plate is large, the wave dissipation effect is weak. Reducing the opening rate can improve the wave dissipation effect, but only to a certain extent. Under irregular wave actions, the velocity field of the submerged horizontal plate is uniformly distributed. The relative submerged depth has minimal effect on the maximum flow velocity and root mean square flow velocity. Increasing the wave height and increasing the open holes on a plate can improve the flow velocity at the bottom of the plate. However, increasing the opening ratio also leads to insignificant changes in flow velocity. A correlation between the transmission coefficient of the open plate and the maximum flow velocity has also been determined. The findings of this paper serve as a research foundation for the implementation of submerged horizontal plate wave dissipation structures in engineering.

**Keywords:** irregular waves; submerged horizontal plate; physical model trial; wave dissipation effect; flow velocity

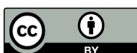

## 1. Introduction

As a novel permeable wave suppression structure [1], the submerged horizontal plate [2,3] offers several advantages, such as a simple structure, material efficiency, low foundation requirements, and unobstructed water circulation. It can effectively mitigate the impact of waves on marine structures [4] and thus meets the needs of modern deep-sea marine engineering, with good application prospects.

In contrast, traditional breakwaters, such as the vertical, inclined, hybrid and floating ones [5], have several drawbacks such as structures being too complex, higher material consumption, higher costs, not being suitable for deep-water operation, difficult to construct, and lengthy construction times.

The submerged horizontal slab is a common simplified version for coastal and marine engineering structures. Heins [6] introduced the theory of water waves in a finite depth channel with a submerged horizontal plate. He pioneered the application of linear

wave theory to analyze wave scattering from such plates. This led to the subsequent application of more cost-effective solutions like plate breakwaters that can reduce the impact of nearshore currents and sediment transport.

Physical model tests conducted by Stoker and Lindsay [7] investigated the hydrodynamic properties of water plates under wave action. They derived formulas for the transmission and reflection coefficients under long-period wave action. Rey and Touboul [8] conducted an experimental study on the protective effect of submerged horizontal plates on coastal areas. They investigated the effect of coastal currents on the reflectivity of submerged horizontal plates and the hydrodynamic loads applied to the plates, under monochromatic and irregular wave conditions. AlYousif et al. [9] conducted experiments to investigate the wave forces and moments acting in the direction of the vertical wall (VW) and the VW attached to the horizontal plate (VWHP). They examined the relationship between wave force and moment patterns to derive the phase lag relationships between horizontal and vertical wave forces for regular waves, and the probability density of random waves. Dong et al. [10] investigated experimentally the forces of monochromatic and isolated waves on submerged horizontal plates in a wave tank and discussed the effect of an uneven bottom on wave loads. Huang et al. [11] provides a novel approach that combines computational fluid dynamics (CFD) with computational solid mechanics (CSM) to dynamically simulate the fully coupled hydroelastic interaction between nonlinear ocean waves and a submerged horizontal plate breakwater (SHPB). The wave attenuation effect is found to be maximized when an SHPB has a deformation amplitude close to the incident wave amplitude. Gao et al. [12,13] analyzed the mechanism of periodic submerged embankments to slow down port resonance and the impact of Bragg reflection. Stachurska et al. [14] investigated the interaction of gravity waves with a semi-submerged rectangular cylinder of elastic bottom, and wave transmission at the structure, wave-induced pressures excreted on the plate and plate deflections were discussed.

Linear wave theory [6,15] was used as numerical modeling to obtain the reflection and transmission properties of monochromatic waves. This analytical and numerical method was applied to analyze wave scattering from a submerged horizontal plate. Carter et al. [16] investigated the inverse flow of a submerged horizontal plate under the action of surface waves in finite water depths based on linear potential theory. Using the boundary element method, they compared the results of the linear potential flow model with those from a non-linear viscous flow model. The mechanism of generation of oscillatory backflow at the bottom of the plate was explained in detail and its fluid nonlinearity and viscosity were analyzed. Yu et al. [17] proposed initial conditions for wave breaking in submerged horizontal plates, based on an empirical formulation of partial standing wave breaking in uniform water depth. They investigated the linear and non-linear wave forces acting on the submerged horizontal plates using the eigenfunction expansion method. Choudhary et al. [18] investigates the scattering of oblique incident waves by two floating horizontal porous plates to study the role of different pairs of barriers in dissipating the incident wave energy.

Using a combination of physical model tests and numerical simulations, Graw [19] employed laser beam deflection and ultrasonic 3D probes to measure the spatio-temporal characteristics of the flow field at the bottom of the plate. The study provided a preliminary explanation of the mechanism that generates oscillatory backflow at the bottom of the plate. It introduced a new concept for a submerged horizontal plate-type wave energy converter. Seiffert et al. [20] demonstrated the results of horizontal and vertical forces due to isolated waves acting on a 2D horizontal plate by performing a series of laboratory experiments as well as CFD calculations. Cheng et al. [21] studied the interaction of focused waves with uniform flow and submerged horizontal plates, They employed a fully non-linear numerical wave flume based on the time-domain high-order boundary element method. This required application of the two-dimensional non-linear potential theory for numerical solutions, followed by experimental validation of these solutions in a two-dimensional glass curtain wall pool. Xu et al. [22] used the meshless particle CFD solver

MLParticle SJTU to numerically simulate the interaction of a two-dimensional submerged fixed horizontal rigid plate with isolated waves and experimentally validated the numerical simulation results. The analyzed results of wave height, inundation depth and plate length were then used to further investigate the relationship between the wave-structure interactions. Gao et al. [23,24] studied fluid resonance inside a narrow gap between two side-by-side boxes breakwaters based on an open-source CFD package OpenFOAM. Seibt et al. [25] performed performed a full scale numerical assessment of the design of the Submerged Horizontal Plate device with the aim of improving its performance. Hayatdavoodi et al. [26] studied the wave-induced oscillations of submerged horizontal plates, and found that the oscillation varies almost linearly with the wave height, but nonlinearly with the wave period, initial submergence depth of the plate, damping, and the spring stiffness.

Currently, research on the permeable submerged horizontal plates has made certain progress, but the focus has mainly been on the interaction between regular waves and horizontal plates of different structural forms. There is relatively less research on the characteristics of irregular waves and the flow field at the bottom of horizontal plates. Therefore, in order to better fit actual engineering applications, after introducing the experimental conditions, this article studies the wave field around submerged horizontal plates under the action of irregular waves and their wave reduction effects. Additionally, the distribution characteristics of the flow field at the bottom of horizontal plates are analyzed, with the aim of providing reference for subsequent research and practical engineering.

## 2. Testing Conditions

### 2.1. Experimental Arrangement

The model tests were carried out in a water tank belonging to the Transportation Department of Tianjin Research Institute for Water Transport Engineering. The tank dimensions were 68 m long, 1 m wide and 1 m deep. It was equipped with a pusher-type wave maker at one end and a wave elimination frame at the other end. The latter serves to absorb wave energy and reduce wave reflection, effectively eliminating the influence of reflected waves on the tests.

The horizontal board was placed in the water tank at about 34 m from the pushing plate of the wave maker. For the test, a single-layer horizontal board model made of synthetic hardwood board was used, which was 1 m long, 1 m wide, and 0.1 m thick. For perforated plates, there are 16 circular holes which setting of 4 × 4 with a spacing of 0.25 m. In the experiment, the opening rate (K = the area of the holes/the area the plate) changes by adjusting the radius of the circular holes (0.05 m, 0.063 m, 0.071 m). The sketch of the perforated plate is shown in Figure 1.

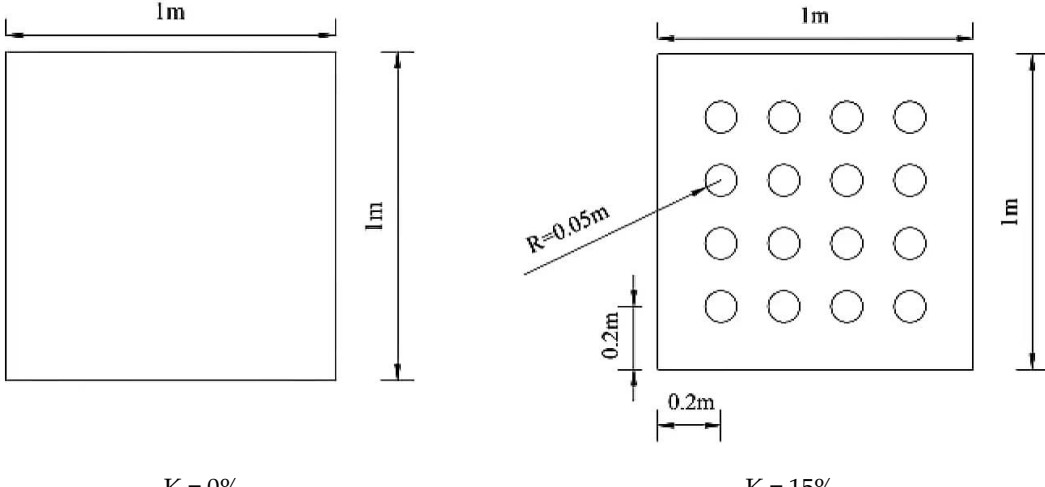

K = 0%                                    K = 15%

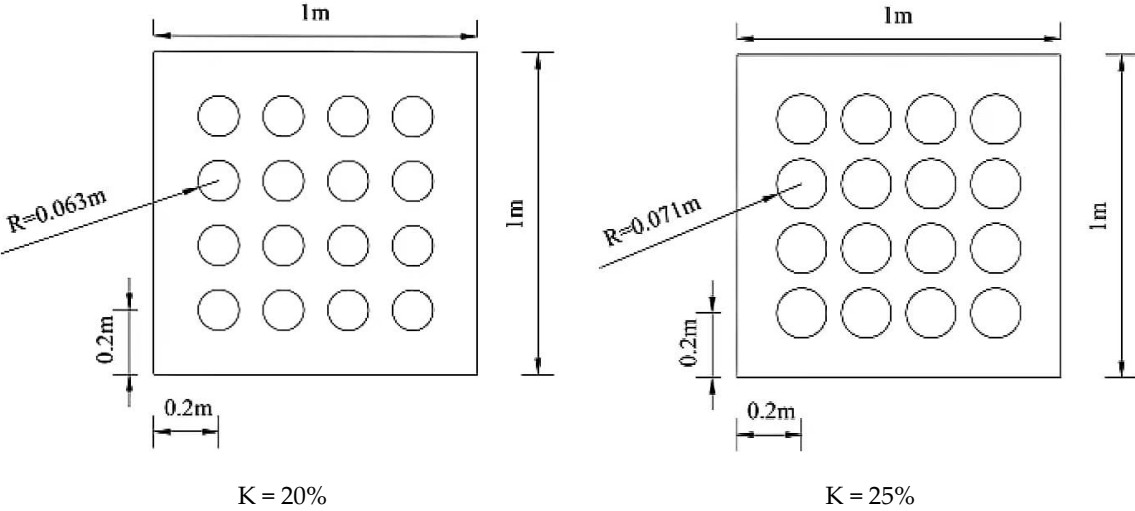

**Figure 1.** Sketch of the perforated plate.

Six wave height meters ($W_1$, $W_2$, $W_3$, $W_4$, $W_5$, $W_6$) were positioned before and after the horizontal board to measure the wave heights before and after the horizontal board. Four Nortek Vectrino Profiler meters ($V_1$, $V_2$, $V_3$, $V_4$) were also arranged at the bottom of the horizontal plate to measure the velocity distribution at the bottom of the plate. The layout of the model and the relative positions of the wave height meters and velocity meters in the water tank are shown in Figure 2.

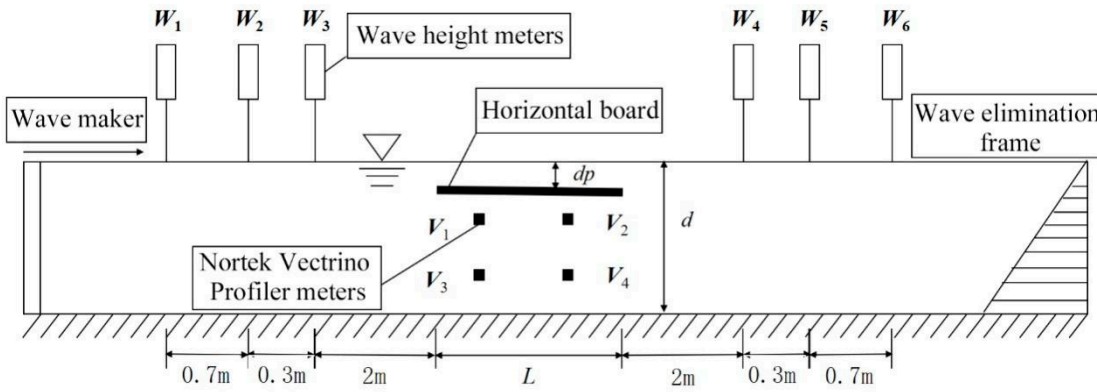

**Figure 2.** Sketch of arrangement of test model in the wave flume.

*2.2. Test Parameters and Groups*

The main experimental physical parameters involved in this thesis are summarized in Table 1. These parameters include water depth (d), length of the horizontal plate along the wave travel direction (L), submergence depth of the horizontal plate ($d_P$), which is the vertical distance between the static water surface and the upper surface of the plate; the Significant wave height of the irregular wave ($H_s$), the significant wave period ($T_s$), the significant wavelength ($\lambda_s$), the reflection coefficient ($C_R$), which equals the reflction wave height/incident wave height, the transmission coefficient ($C_T$), which is the transmission wave height/incident wave height, and the spectral function (S (f)) (as shown in Figure 1).

**Table 1.** Main experimental parameters.

| Main Parameters | Indication Symbols | Units | Main Parameters | Indication Symbols | Units |
|---|---|---|---|---|---|
| Testing water depth | d | m | Opening ratio | K | / |
| Horizontal plate submergence depth | $d_P$ | m | Effective wavelength | $\lambda_s$ | m |
| Length of horizontal plate | L | m | Wave frequency | f | Hz |
| Significant wave period | $T_s$ | s | Reflection coefficient | $C_R$ | / |
| Significant wave height | $H_s$ | m | Relative dive depth | $d_P/d$ | / |
| Relative plate length | $L/\lambda_s$ | / | | | |

For the application of submerged horizontal plates in sea conditions, the design prototype was set at a water depth of 6.4 m, and the test water depth at 0.4 m, in accordance with the gravity similarity criterion, and the geometric scale for the experiment is 1; 16. The tests were conducted using irregular waves, and tested on models with different opening rates and submerged depths. The wave elements and horizontal plate dimensions of the prototype and model are presented in Table 2 (Three "Hs" correspond to a "d", and nine "Ts" correspond to each "Hs") and Table 3 respectively. The test groups are presented in Table 4 (Nine "Ts" correspond to "Hs" of 0.05 and 0.1, five "Ts" correspond to "Hs" of 0.15).

The test was replicated thrice for each wave element and the average value was taken as the final test result. The wave height and velocity were acquired at intervals of 0.02 s. and the sampling time length was adjusted according to the working conditions. In the experiment, the waves superimposed by the incident wave and reflected wave are separated using the two point separation method of Goda [27]. Through comparison, it was found that the results obtained by the two point method can basically meet the accuracy requirements of this experiment. According to the requirements of this method, the distance between adjacent wave altimeters must meet the following requirements:

$$0.05L_{max} < \Delta l < 0.45L_{min} \tag{1}$$

where $L_{max}$ and $L_{min}$ are the longest and shortest wavelengths among all wave elements in the experiment, and it takes $\Delta l = 0.3$ m in the paper.

**Table 2.** Wave element.

| Prototypes | | | Models | | |
|---|---|---|---|---|---|
| **h/m** | **Ts/s** | **Hs/m** | **h/m** | **Ts/s** | **Hs/m** |
| 6.4 | 10.68, 7.32, 5.8, 4.92, 4.36, 3.96, 3.64, 3.4, 3.24 | 0.8, 1.6, 2.4 | 0.4 | 2.67, 1.83, 1.45, 1.23, 1.09, 0.99, 0.91, 0.85, 0.81 | 0.05, 0.1, 0.15 |

**Table 3.** Horizontal plate size.

| | Length/m | Broad/m | Thick/m | Submergence Depth/m |
|---|---|---|---|---|
| Prototype | 16 | 16 | 1.6 | 2.24, 1.6, 0.96, 0.32 |
| Model | 1 | 1 | 0.1 | 0.14, 0.1, 0.06, 0.02 |

**Table 4.** Model test cases.

| Depth d/m | Submergence Depth dp/m | Opening Ratio | Irregular Wave Effective Wave | |
|---|---|---|---|---|
| | | | Effective Period Ts/s | Height Hs/m |
| 0.4 | 0.14, 0.1, 0.06, 0.02 | 0 | 2.67, 1.83, 1.45, 1.23, 1.09, 0.99, 0.91, 0.85, 0.81 | 0.05, 0.1 |
| | | | 2.67, 1.83, 1.45, 1.23, 1.09 | 0.15 |
| 0.4 | 0.1, 0.06, 0.02 | 0.1, 0.15, 0.2 | 2.67, 1.83, 1.45, 1.23, 1.09, 0.99, 0.91, 0.85, 0.81 | 0.05, 0.1 |
| | | | 2.67, 1.83, 1.45, 1.23, 1.09 | 0.15 |

## 3. Effect on Transmitted Wave Period

The submerged horizontal plate is used as a wave dissipation structure in engineering, and its attenuation effect on the surrounding wave period and wave height are key concerns. In this chapter, the variation law of the transmitted wave period on the back wave side of the horizontal plate is discussed in relation to two parameters: relative submergence depth (inundation depth/water depth) and open aperture rate. To analyze the characteristics of the transmitted wave period, the period ratio, defined as the ratio of the effective period of transmitted waves to the effective period of incident waves ($T_t/T_P$), is used as the vertical coordinate. The relative plate length, expressed as the ratio of the plate length to the effective wavelength ($L/\lambda_s$), is used as the horizontal coordinate. These two coordinates are used to plot the variation curve.

### 3.1. Effect of Relative Dive Depth on the Cycle

This section focuses on examining the effect of relative dive depth ($d_P/d$) on the wave period. The results are compared with the same opening rate and significant wave height, with different relative dive depths by varying the relative plate lengths. The data from three wave height meters: $W_4$ (before), $W_5$ (middle) and $W_6$ (after), were extracted for comparison after each wave condition. For clarity, the applicable period of irregular waves is referred to as "period", the significant wave height of irregular waves as "wave height", and the effective wavelength of irregular waves as "wavelength". It is worth noting that the relative submergence depth (inundation depth/water depth) and the fixed water depth of this test only differ by a coefficient. For ease of understanding, the more intuitive concept of submerged water depth is used for explanation in this section instead of relative submergence depth.

The results of the period ratio for the 20% open-aperture plate, under the working conditions with significant wave heights, $H_s = 0.1$ m and $H_s = 0.15$ m, are presented in Figure 3. Here, the transmission period Tt is obtained using the average results of the three wave height meters located after the dike. In order to illustrate the differences in the observation results of the three wave altimeters, the relative average error amplitudes ($\frac{\sum|T_i-\overline{T}|}{3}/\overline{T}$) of the periodic results of the three observation points behind the embankment under different working conditions were calculated. Figure 3a shows that the relative average error amplitudes under different wave conditions are between 0.42% and 7.78%, while Figure 3b shows that the relative average error amplitudes under different wave conditions are between 0.46% and 6.16%. It was found that the transmission period decreases as the relative submergence decreases. The attenuation of the transmittance period, especially for shorter waves (with larger relative plate lengths), was significant. This indicates that the closer the perforated plate is to the water surface, the more pronounced the attenuation of the transmittance period becomes. The reason is that the closer the horizontal plate is to the water surface, the greater the blocking effect on the upper layer wave energy, thus having a significant impact on the period.

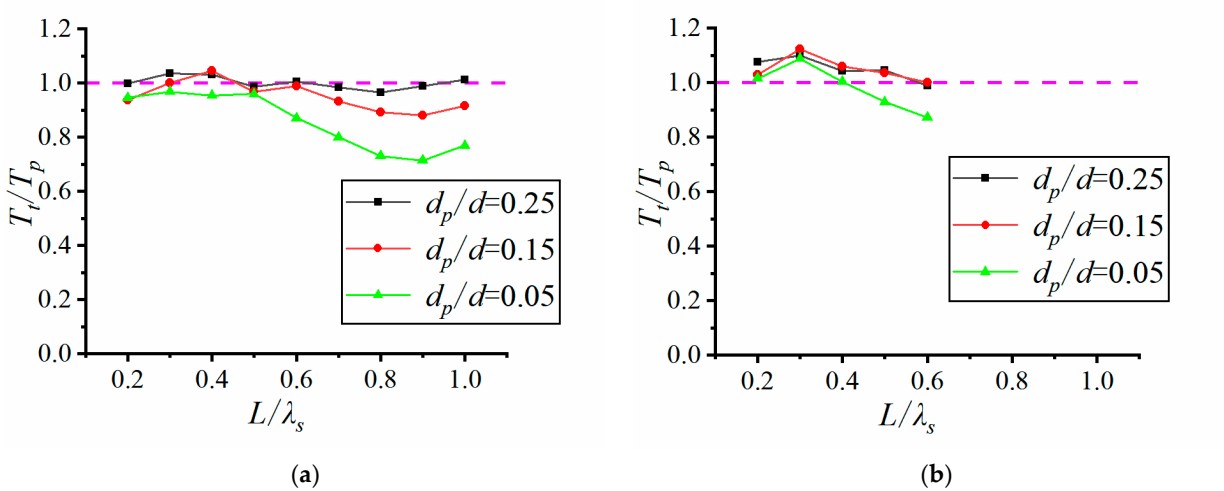

**Figure 3.** The comparisons of wave period ratio of 20% opening perforated plate ($H_s$ = 0.1 m, $H_s$ = 0.15 m). (**a**) Significant wave height $H_s$ = 0.1 m; (**b**) Significant wave height $H_s$ = 0.15 m.

*3.2. Effect of Opening Rate on Cycle Time*

This section examines the effect of the open aperture ratio on the wave period. The variation curve of the period ratio with relative plate length is plotted using the period ratio $T_t/T_P$ (transmitted wave effective period/incident wave effective period) as the vertical coordinate and the relative plate length $L/\lambda_s$ (plate length/effective wavelength) as the horizontal coordinate. The results are compared for each relative dive depth, with the same significant wave height but different opening ratios.

Figure 4 illustrates the variation of the transmission period of horizontal plates with different open aperture ratios for three relative dive depths with an significant wave height of 0.05 m. After calculation, the average error amplitude variation ranges for each wave condition in Figure 4a–c are 0.53~3.56%, 0.52~14.73%, and 0.32~13.72%, respectively.

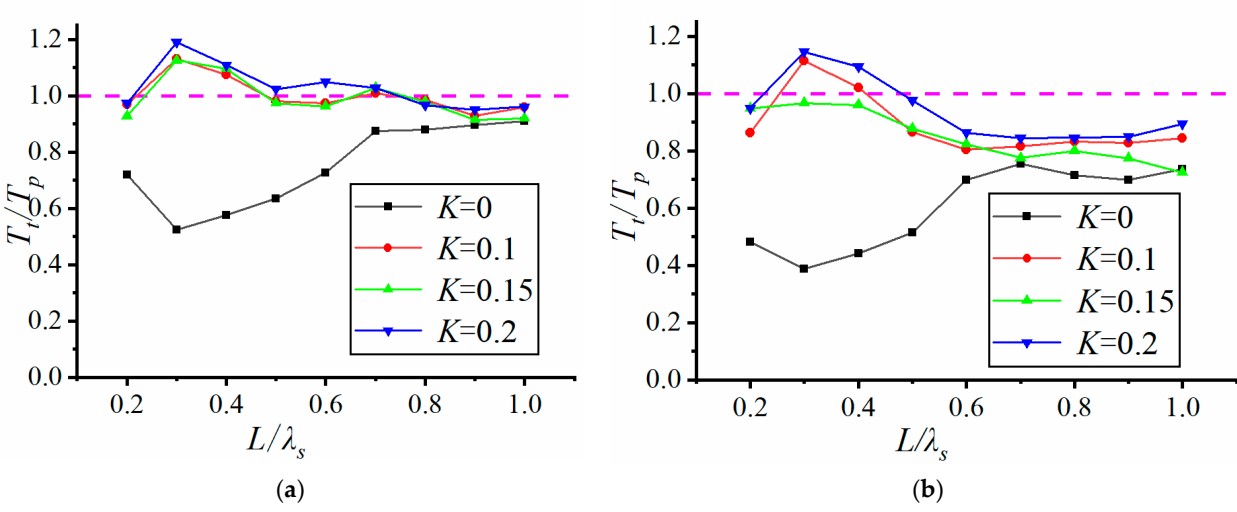

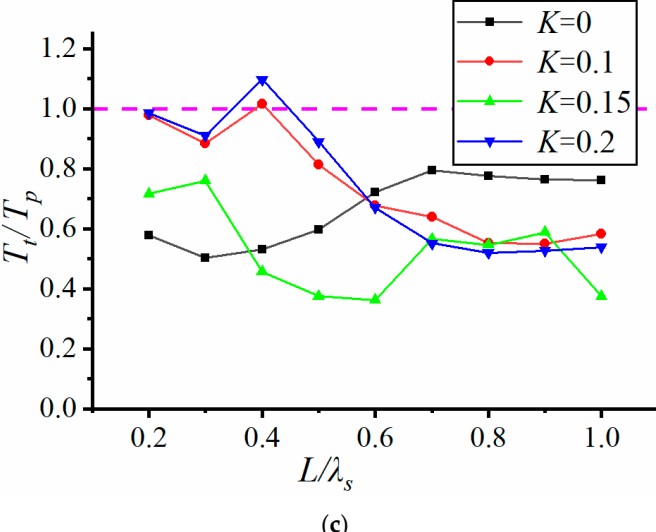

**(c)**

**Figure 4.** The comparisons of wave period ratio of significant wave height Hs = 0.05 m. (**a**) Relative submerged depth $d_P/d = 0.25$; (**b**) Relative submerged depth $d_P/d = 0.15$; (**c**) Relative submerged depth $d_P/d = 0.05$.

The trends of each curve in Figure 4 show that the period ratio of the solid plate tends to decrease initially and then increase as the relative plate length increases, while the period ratio of the perforated plate tends to increase before decreasing. The inflection point of the curve generally appears in the range of 0.3–0.4 value of the relative plate length, suggesting that the ratio of wavelength to plate length can affect the trend of the Significant wave period.

From Figure 4a, it is evident that when relative diving depth, $d_P/d = 0.25$, the period ratio of the long-period perforated plate is greater than 1.0, and that of the short-period plate is close to 1.0. Comparing the effect of the opening rate, it can be noted that the curves of the three perforated plates are almost identical. In addition, the period ratio of the solid plate in the long-wave (small relative plate length) case is significantly smaller than that of the perforated plate. while it is significantly higher in the short-wave case. The period ratios of the solid plate and perforated plate are also more consistent in the short-wave case. This indicates that the horizontal plate without an open hole can significantly reduce the transmitted wave period. But once the hole is activated, the wave period decay is not much related to the opening rate.

From Figure 4b, when the relative dive depth, $d_P/d = 0.15$, the period ratio is lower than 1.0 for most wave conditions (short wave case with larger relative length). On opening the hole, the curves became roughly similar under short wave conditions, and the solid plate made the period decrease significantly under long wave conditions.

Figure 4c reveals that when using a 15% perforated plate with relative $d_P/d = 0.05$, the period ratio reaches its minimum. The attenuation of the transmission wave period is most significant, and the cycle ratio of some short waves (relative to the plate length is larger) is even smaller than that of the solid plate. By analyzing the curves for different opening ratios, it becomes evident that as the opening ratio decreases, the attenuation of the transmitted period initially increases and then decreases, with the inflection point occurring at an opening ratio of 15% under test conditions. In this paper, the perforation rate falls within the moderate range of 15%, and the most significant attenuation of the transmitted wave period is when the submergence depth is small. This phenomenon suggests that a judicious choice of opening ratio (neither too large nor too small) can optimize the period attenuation.

## 4. Effect on Transmitted Wave Height

This chapter primarily discusses the effect the submerged horizontal plate exerts on the wave height. Lower permeable wave dissipation structure usually uses transmission coefficient $C_T$ (transmitted wave height/incident wave height) and reflection coefficient $C_R$ (reflected wave height/incident wave height) to study the variation of wave height around the structure. Since wave energy is proportional to the square of the wave height, this section mainly examines the transmission coefficient of the submerged horizontal plate. The effects of wave height, relative submergence and opening rate on wave height and wave energy are also discussed.

### 4.1. Effect of Wave Height on Transmission Coefficient

Significant wave height is crucial in assessing wave dissipation performance. This section will thus focus on discussing the relationships between significant wave height, transmission coefficient and reflection coefficient.

Figure 5 shows the variation curves of the transmission coefficient with their respective relative plate lengths for four different relative submergence conditions. The four corresponding relative dive depths are 0.14 m, 0.1 m, 0.06 m and 0.02 m, respectively. The relative average error amplitudes ($\frac{\sum|H_i - \overline{H}|}{3}/\overline{H}$) of the wave height results at the three observation points behind the embankment were also calculated. The relative average error amplitudes in Figure 5a–d ranged from 10.26~41.06%, 16.47~47.16%, 21.17~62.74%, and 28.04~66.67%, respectively. The relative error amplitude of this group varies greatly due to the different incident wave heights of each wave condition, which can lead to significant changes in wave heights at different positions behind the embankment. Although other groups (such as Figures 6 and 7) also calculate the relative error amplitude of wave height measurement results behind the embankment, their corresponding working conditions are the same incident wave heights, so the results of the three wave heights behind the embankment have a small difference.

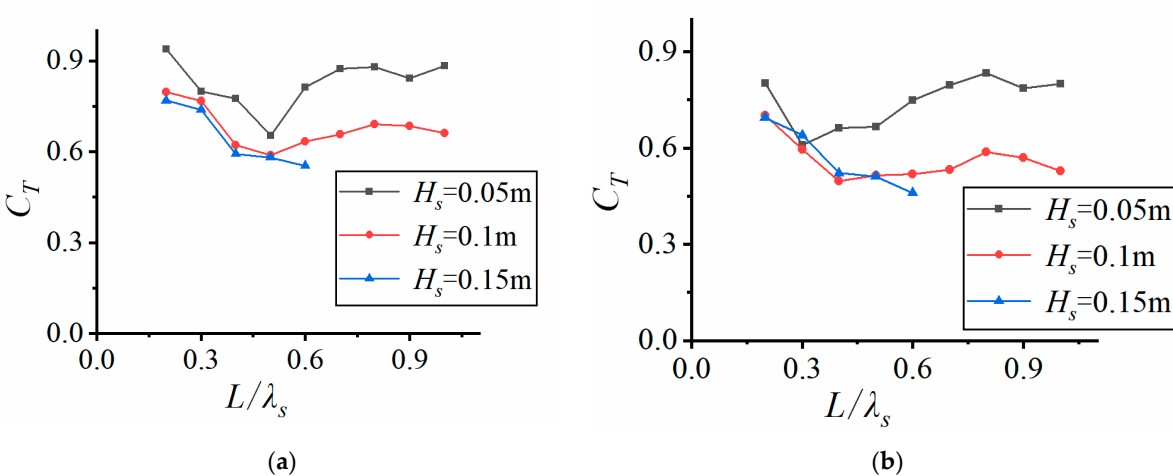

(**a**)                                                                 (**b**)

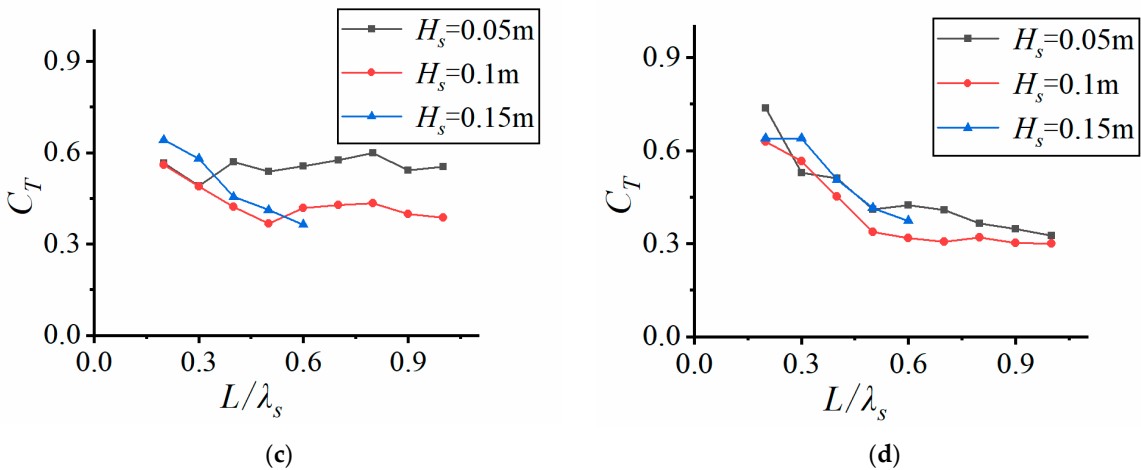

(**c**)  (**d**)

**Figure 5.** Effect of significant wave height on transmission coefficient at different relativesubmerged depth. (**a**) Relative submerged depth $d_P/d = 0.35$; (**b**) Relative submerged depth $d_P/d = 0.25$; (**c**) Relative submerged depth $d_P/d = 0.15$; (**d**) Relative submerged depth $d_P/d = 0.05$.

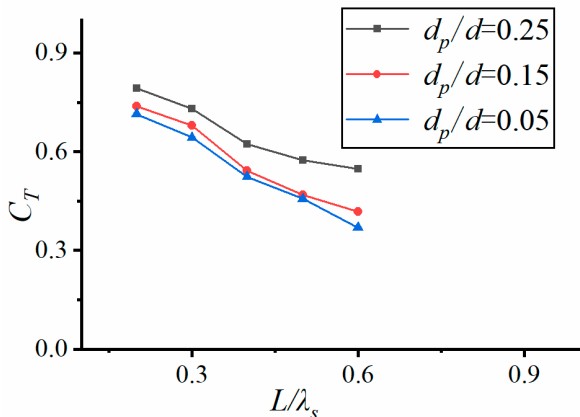

**Figure 6.** Transmission coefficient, reflection coefficient and energy dissipation of perforated plate ($K = 0.1$, $H_s = 0.15$ m).

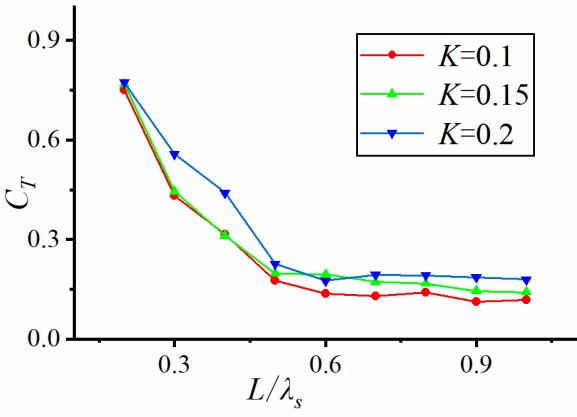

**Figure 7.** Effect of porosity on transmission coefficient, reflection coefficient and energy dissipation ($d_P/d = 0.05$, $H_s = 0.05$ m).

For the conditions depicted in Figure 5a,b, the inundation depth is greater than the effective wave amplitude (half of the wave height). It is evident that the larger the significant wave height is, the smaller the transmission coefficient, indicating that the wave dissipation effect is better in the large wave case. In Figure 5c,d, the submergence depths are less than the effective wave amplitudes, which means that there were more wave-breaking phenomena. In other words, the effect of significant wave height on the transmission coefficient is minimal. It can also be observed that the relative submergence depth (inundation depth) exerts a greater influence on the wave dissipation effect. This will be further explored in the subsequent sections.

*4.2. Effect of Relative Dive Depth on Transmission Coefficient*

An experimental study was conducted on the horizontal plate with an opening rate K = 10% at three different wave heights and three inundation depths. The significant wave height $H_s$ = 0.15 m is set, which is considered a relatively large value. It corresponds to five groups of periods (see Table 4 for details). Figure 6 shows the variation curves of transmission coefficient $C_T$, reflection coefficient $C_R$ and energy dissipation with the relative plate length at three relative diving depths (submergence depth) of the horizontal plate. Since the water depth d was fixed as a constant in the tests, the corresponding inundation depths for the three relative dive depths ($d_P/d$ = 0.25, 0.15, 0.05) are 0.1 m, 0.06 m and 0.02 m, respectively. After calculation, the relative average error amplitude of each wave condition in Figure 6 is between 1.41% and 8.52%.

As seen from Figure 6, increasing the horizontal plate opening rate to 10% leads to a decrease in the transmission coefficient as the relative plate length increases. However, the reflection coefficient does not change significantly with respect to the relative plate length, while the energy dissipation increases as the relative plate length increases. Since the wave height of this wave condition is larger and the short-period condition is not tested, the result curve change law is relatively simple and clear.

Comparing the curves for the three relative submergence depths in Figure 6, we can see that when the submergence depth (relative submergence depth) increases, the transmission coefficient increases while the reflection coefficient and the energy dissipation decrease. This can be attributed to the fact that for a perforated horizontal plate, when the submergence depth is large, the significant wave height is relatively small. The waves propagate from above the plate, and the blocking effect of the horizontal plate and the open hole is smaller. The result is a high transmitted wave, a low reflected wave, and low energy dissipation with most of the waves carrying the wave energy above the plate. For the two cases with smaller inundation depths (0.06 m and 0.02 m), the significant wave height is 0.075 m, which is higher than the two inundation depths, so most waves bottom out after passing through the horizontal plate. The hole on the plate intensifies the breaking phenomenon and consumes the wave energy. Hence, the transmission coefficient curve and wave energy dissipation curve are similar in these two wave cases.

*4.3. Effect of Open Ratio on Transmittance Coefficient*

The aim of this paper is to increase the wave dissipation effect by creating open holes in the horizontal plate to vary its permeability. This section mainly discusses the effect of the opening rate on the transmission coefficient, reflection coefficient and energy dissipation of the horizontal plate. The tests were conducted at three dive depths, three wave heights and three opening rates of the horizontal plate. The relative dive depth was fixed at $d_P/d$ = 0.05.

Figure 7 presents the variation curves of the transmission coefficient, reflection coefficient and energy dissipation with relative plate length and three different open apertures, corresponding to an significant wave height of $H_s$ = 0.05 m. After calculation, the relative average error amplitude of each wave condition in Figure 7 is between 0.61% and 12.95%.

As seen in Figure 6, when the open aperture rate is 0.2, the transmission coefficient and the reflection coefficient became higher, the energy consumption is lowered, and the wave dissipation performance is generally weaker. This phenomenon shows that increasing the aperture area does not necessarily enhance the wave dissipation performance. This is because as the aperture increases, the waves are more likely to pass through the horizontal plate, and the wave breakage decreases, which leads to the increase of transmission coefficient and the weaker wave dissipation effect.

It can also be seen that the curves of the three opening ratios are relatively close when the relative plate length is large (greater than 0.5). This indicates that for short waves (effective wavelength less than 2 times the horizontal plate length), changing the opening rate does not have a significant effect on the wave dissipation performance.

The study of the flow field at the bottom of a horizontal plate not only provides a comprehensive understanding of the blocking effect of this structure on waves and currents, but also provides a research basis for setting up high-efficiency ocean energy conversion devices near the horizontal plate.

## 5. Effect on Flow Rate

The submerged horizontal plate is part of the permeable wave dissipation structure, with water flowing through the lower part to facilitate water exchange. This permeability is key to this structure and the reason why such a structure has been strongly promoted in recent years. More attention has been given to the wave dissipation performance of the horizontal plate, with little research on the bottom flow field. This section aims to conduct a preliminary study of the flow field at the bottom of the horizontal plate, to explore the flow velocity at specific points in the lower part of the solid plate and the overall distribution of the flow field. In addition, it further examines the influence of relative diving depth and opening rate on flow velocity.

### 5.1. Measurement and Uniformity Analysis of Plate Bottom Velocity Field

In the experiment, the flow velocity of the four measuring points under the board (corresponding to $V_1$, $V_2$, $V_3$, and $V_4$) at each time point was measured. Based on this, the flow velocity time history curve in the time domain was plotted, and the time domain curve was transformed into the frequency domain velocity spectrum curve through Fourier transform. By comparing the velocity time history curves and velocity spectra of the four measurement points in the X-direction, it was found that the shapes of the velocity time history curves and velocity spectra of the four measurement points were basically the same, and the positions of the four spectral peaks and peaks were basically the same. This can prove that the X-direction velocity of each point at the bottom of the horizontal plate is close. In the experiment, it was found that the Z-direction velocity is generally in the order of 0.01 m/s, which is relatively small compared to the X-direction velocity. Therefore, it can be said that the distribution of the flow field at the bottom of the horizontal plate is relatively uniform. One point velocity can be used as a representative for research. The $V_2$ measurement point results will be used for analysis in the following text.

In order to study the quantitative characteristics of flow velocity, the root mean square flow velocity (calculated as Formula (2)) was used for discussion.

$$V_{rms} = \sqrt{\frac{\sum_{i=1}^{N} V_i^2}{N}} = \sqrt{\frac{V_1^2 + V_2^2 + ... + V_N^2}{N}} \tag{2}$$

There, $N$ represents a total of $N$ flow velocity data, $V_i$ represents the instantaneous flow velocity at each time, and $V_{rms}$ represents the root mean square flow velocity.

Table 5 lists the RMS (Root Mean Square) flow rates at several representative working conditions for the $V_1$–$V_4$ measurement points at the bottom of the solid plate.

**Table 5.** RMS velocity of solid plate bottom measurement point under different working conditions.

| Wave Height m | Period s | Submergence Depth m | Root Mean Square Velocity in X-Direction m/s | | | | Root Mean Square Velocity in Z-Direction m/s | | | |
|---|---|---|---|---|---|---|---|---|---|---|
| | | | $V_1$ | $V_2$ | $V_3$ | $V_4$ | $V_1$ | $V_2$ | $V_3$ | $V_4$ |
| 0.05 | 0.85 | 0.14 | 0.012 | 0.011 | 0.015 | 0.018 | 0.005 | 0.004 | 0.004 | 0.005 |
| 0.05 | 0.99 | 0.1 | 0.022 | 0.024 | 0.018 | 0.023 | 0.010 | 0.009 | 0.011 | 0.010 |
| 0.1 | 1.23 | 0.06 | 0.072 | 0.065 | 0.078 | 0.063 | 0.029 | 0.022 | 0.025 | 0.024 |
| 0.15 | 2.67 | 0.02 | 0.112 | 0.102 | 0.126 | 0.118 | 0.032 | 0.028 | 0.036 | 0.034 |

From Table 5, it can be seen that the root mean square velocity in the X-direction and Z-direction of the $V_1$–$V_4$ measurement points at the bottom of the solid plate are very close under different working conditions, further confirming that the flow velocity at the bottom of the horizontal plate is basically uniformly distributed. In the following discussion, only the root mean square velocity results at the $V_2$ measurement point will be taken for discussion.

*5.2. Effect of Relative Dive Depth on the Flow Rate of Perforated Plate*

This section investigates the effect of relative dive depth on the maximum flow velocity and root-mean-square flow velocity in the X-direction and Z-direction under open-aperture plate conditions. Figure 8 shows the results of plotting the variation of maximum flow velocity with relative plate length for the same significant wave height, 15% open aperture rate and at different relative dive depths. For the 15% perforated plate with significant wave height Hs = 0.1 m, it can be seen that the maximum flow velocities in both the X-direction and Z-direction tend to decrease gradually with the increase in relative plate lengths. The effect of relative submergence on the maximum flow velocity is not significant, which is similar to that of the solid plate, except that the decreasing magnitudes of the X-direction curves increase slightly.

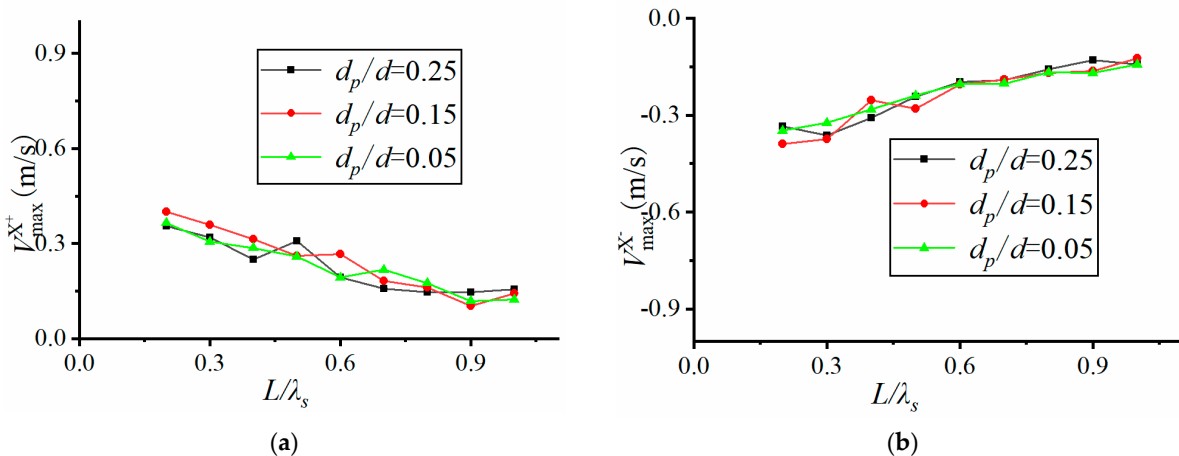

(a)　　　　　　　　　　　　　　　(b)

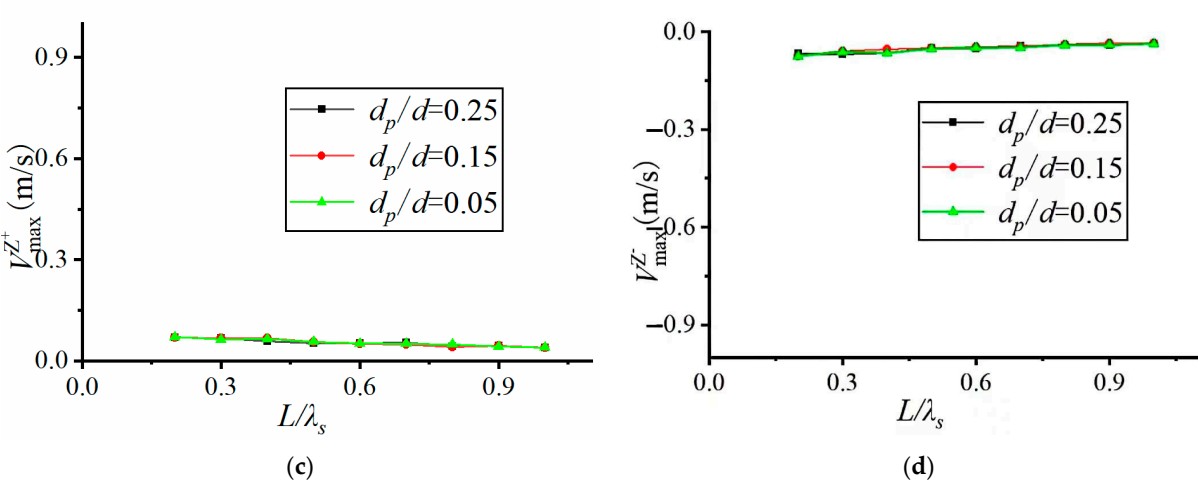

**Figure 8.** Influence of relative plate length on maximum velocity under different relative submerged depths (k = 0.15, Hs = 0.1 m). (**a**) Positive maximum velocity in X direction; (**b**) Negative maximum velocity in X direction; (**c**) Positive maximum velocity in Z direction; (**d**) Negative maximum velocity in Z direction.

*5.3. Effect of Opening Ratio on Flow Rate*

In this section, the effect of the opening rate on the maximum flow velocity and root-mean-square flow velocity in each direction is examined for perforated plate conditions. Figure 9 compares the results of the maximum flow velocity for different relative plate lengths at the same relative depth but with different significant wave heights. When relative depth $d_p/d = 0.25$ and significant wave height Hs = 0.05 m, the flow velocities in X and Z directions exhibit a general trend where maximum flow velocity decreases with an increase in relative plate length, while the Z-direction flow velocity is smaller in value and less varied. Comparing the four sets of curves with different opening ratios, it is evident that the opening ratio can affect the maximum flow velocity. In Figure 8a, comparing the two curves with K = 0 and K = 0.15 shows that the flow velocity of the solid plate (K = 0) is lower than that of the open-hole plate (K = 0.1) in all eight conditions except one. This finding indicates that perforation increases the circulation performance of the upper and lower water bodies, thus increasing the flow velocity in the lower part of the horizontal plate. Comparing these two types of perforated plates (K = 0.1 and K = 0.15), it is noted that the maximum flow velocity increased when the relative plate length was large (smaller wavelength) after the perforated rate was increased. This indicates that increasing the perforated area in the short-wave case can amplify the flow velocity at the bottom of the plate, whereas this effect is not significant in the long-wave cases. Comparing the two types of open-aperture plates, K = 0.1 and K = 0.2, it was found that the two curves were relatively close to each other, indicating that the perturbation effect of open-aperture on the bottom flow velocity becomes less noticeable after increasing the open-aperture ratio to a certain degree.

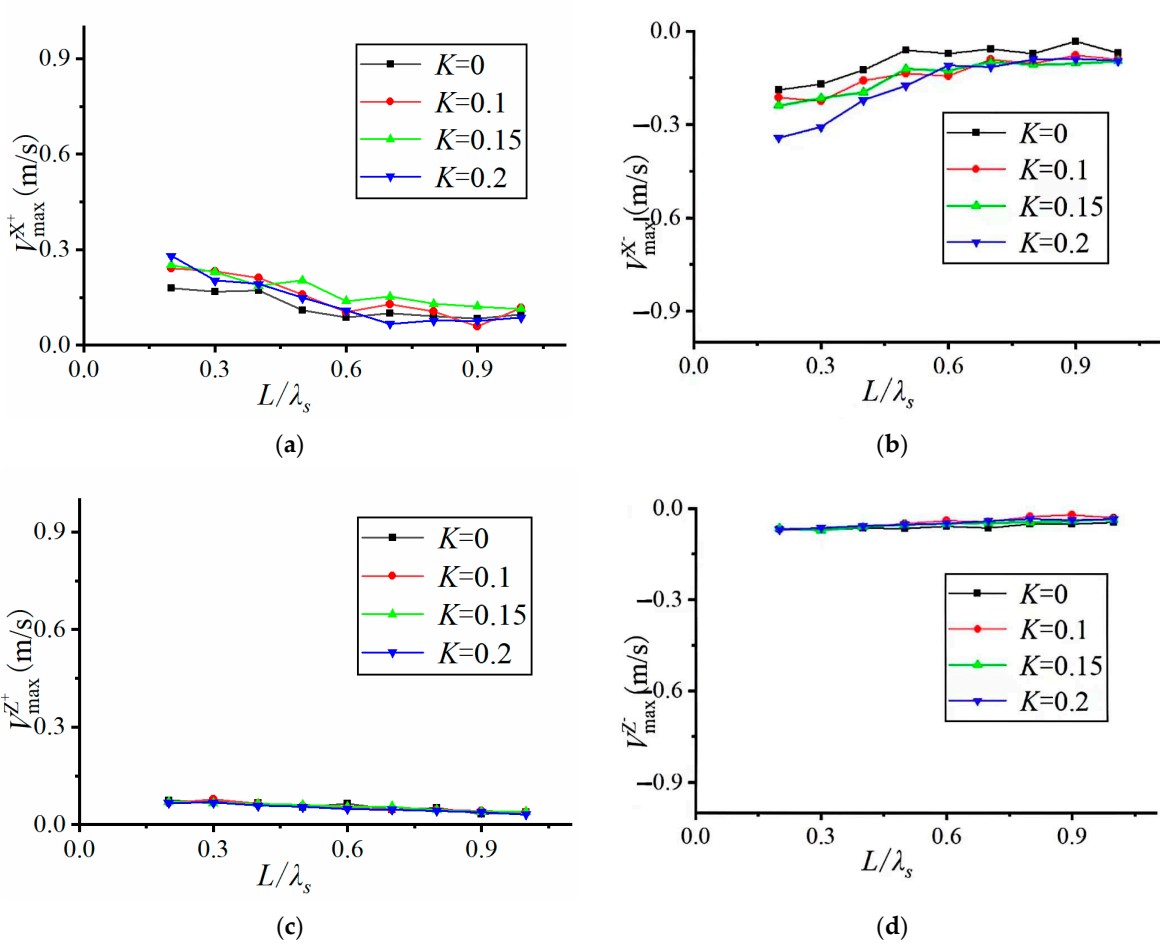

**Figure 9.** The influence of relative plate length on the maximum flow rate under different opening ratios (dp/d = 0.25, $H_s$ = 0.05 m). (**a**) Positive maximum velocity in X direction; (**b**) Negative maximum velocity in X direction; (**c**) Positive maximum velocity in Z direction; (**d**) Negative maximum velocity in Z direction.

Figure 10 shows the test site diagram. When the hole opened, the water exchange occurred at the lower end of the horizontal plate, and many bubbles were generated.

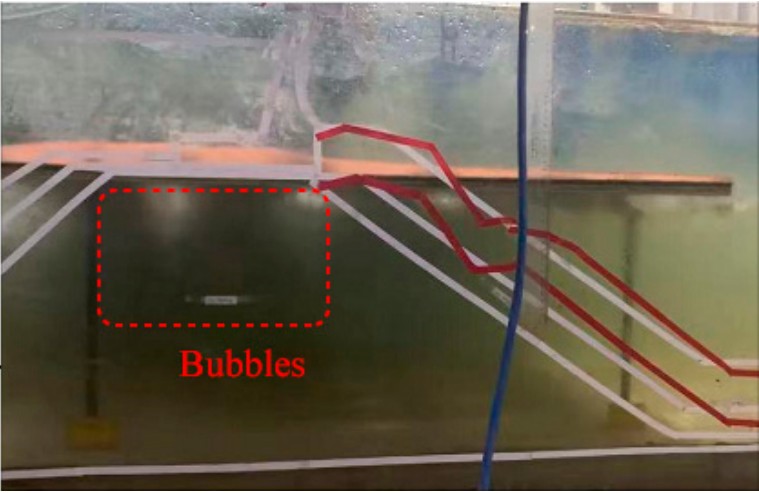

**Figure 10.** Drawing of test site (K = 0.1, $H_s$ = 0.05 m, $d_p/d$ = 0.25).

Subsequently, the effects of the opening rate on the RMS flow significant wave height velocity in the x-direction and z-direction were compared. Figure 11 shows the variation of RMS flow velocity with relative plate length, at the same relative dive depth but different significant wave heights. When relative dive depth $d_P/d = 0.25$ and $H_s = 0.15$ m, it can be seen that the RMS flow velocity in the X-direction and Z-direction shows a gradual decrease with an increase in relative plate lengths, but this change is not significant at a lower velocity in the Z-direction. It is also observed from Figure 11a that the RMS flow velocity of a solid plate is the smallest, especially when the relative plate length (long wave) is small, and the flow velocity increases significantly after the hole is opened. For the open-perforated plate, the three curves almost overlapped, indicating that the effect of the open-perforation rate on the root-mean-square flow velocity is not prominent.

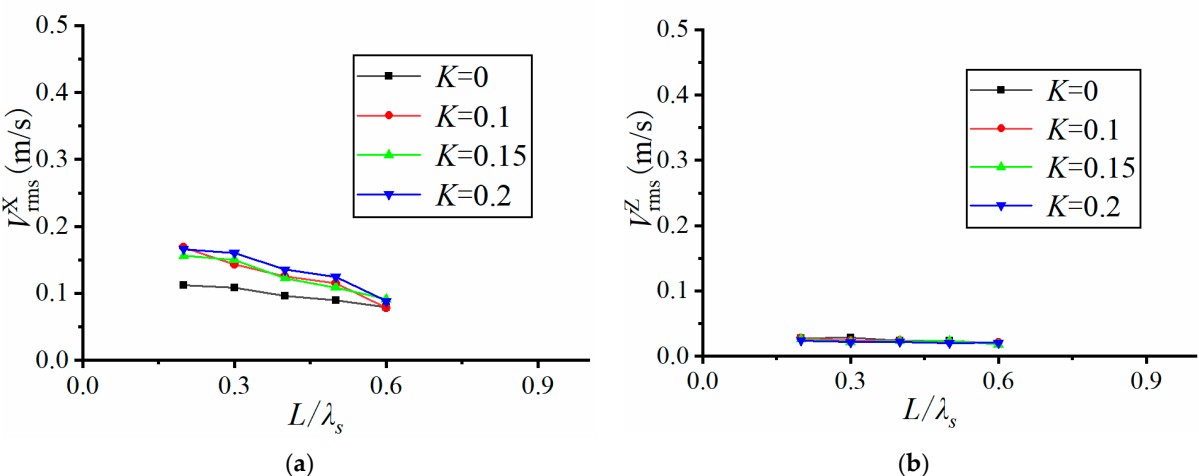

**Figure 11.** Effect of relative plate length on RMS velocity with different opening ratios ($d_P/d = 0.25$, $H_s = 0.15$ m). (**a**) RMS velocity in the X direction; (**b**) RMS velocity in the Z direction.

*5.4. Transmittance Coefficient and Flow Rate*

As a wave dissipation structure, the submerged horizontal plate should have a certain correlation between the wave elimination function in the superstructure and the water permeability function of its lower part. This section mainly focuses on exploring the relationship between the transmission coefficient and the maximum flow velocity. The variation of transmission coefficient and maximum flow velocity in X and Z directions with different relative plate lengths are analyzed for solid plates and 3, the maximum flow velocity and root mean square velocity of solid plate decreases with an increase in relative plate lengths, and increase correspondingly with an increase in significant wave heights. The ratio of maximum flow velocity/root mean square velocity is normally distributed at about 3.0. Whether it is a solid plate or a perforated one, relative submergence has no significant effect on the maximum flow velocity and root mean square velocity in both X and Z directions. The open hole can increase the flow velocity, but with an increased open hole rate, the maximum flow velocity and root mean square velocity of the horizontal plate do not change significantly. The correlation between the transmission coefficient of the solid plate and the maximum velocity at the bottom of the plate is not strong, compared to the correlation with the perforated plate. The higher the incident wave height, the more obvious the correlation becomes.

Figure 12 shows the results for the solid plate, at relative dive depth $d_P/d = 0.35$. Comparing the results for the three wave heights, the variation patterns of the transmission coefficient curve and the maximum flow velocity curve with large wave height ($H_s = 0.15$ m) in Figure 12c shows a general decrease when the relative plate lengths decrease. The discussion mainly focused on the X-direction flow velocity due to insignificant Z-direction

flow velocity. When wave height is small (H$_s$ = 0.05 m, H$_s$ = 0.1 m), the transmission coefficient decreases and then increases, while the flow velocity monotonically decreases. This phenomenon indicates that the correlation between the transmission coefficient and the maximum flow velocity at the bottom of the solid plate is not particularly strong, especially in cases with small wave heights.

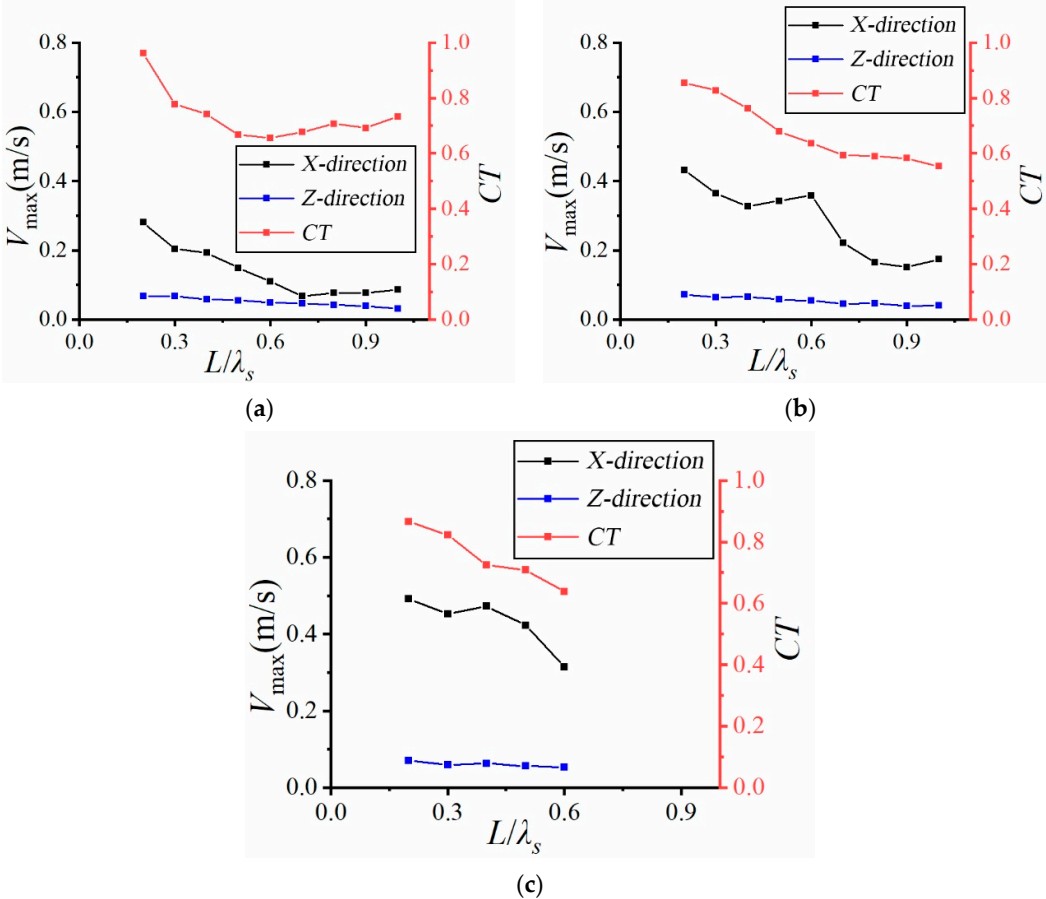

**Figure 12.** Biaxial diagram of perforated plate Velocity-Transmission coefficient (K = 0.2, d$_P$/d = 0.25). (**a**) Significant wave height H$_s$ = 0.05 m; (**b**) Significant wave height H$_s$ = 0.1 m; (**c**) Significant wave height H$_s$ = 0.15 m.

It is evident from Figure 12 that the trend of the transmission coefficient with respect to the relative plate length is similar to that of the maximum flow velocity in the X-direction. This indicates that increasing open holes on the plate can increase the energy exchange between the upper and lower parts of the horizontal plate. The transmission coefficient and the velocity curve exhibit similar changes and demonstrate a strong correlation. Furthermore, this correlation becomes more pronounced with increasing incident wave heights.

## 6. Conclusions

In this paper, the interaction between irregular waves and submerged horizontal plates is analyzed, based on the test results of each working condition in this experiment, leading to the following main conclusions.

Solid submerged horizontal plates attenuate wave periods. The farther away the wave is from the horizontal plate, the greater the attenuation of the wave period. The attenuation of the wave period is more significant when the submerged depth is small. Selecting an appropriate opening rate can optimize the period attenuation.

The larger the submerged depth, the higher the transmission coefficient and the lower the reflection coefficient and energy dissipation, resulting in a poorer wave elimination effect. Reducing the opening rate can improve this wave elimination effect, but only to a certain extent.

The maximum flow velocity and root mean square velocity of solid plate decreases with increasing relative plate lengths and significant wave heights. The relative submergence has no significant effect on the flow velocity. The presence of perforations can increase the flow velocity. However, it does not affect the maximum flow velocity and root mean square velocity of the horizontal plate significantly. The correlation between the transmission coefficient of the solid plate and the maximum velocity at the bottom of the plate is not as strong, compared to that of the perforated plate.

Further research can be conducted in the following areas in the future. For perforated horizontal plates, further discussion can be conducted on the influence of factors such as the position, shape, and size of the holes on various coefficients and energy in the future. The stress situation of the horizontal plate, especially at the connection point of the fixed horizontal plate, is also a work that can be continued in the future.

**Author Contributions:** Conceptualization, Y.M.; Methodology, Y.Z. (Yanna Zheng); Software, Y.Z. (Yifan Zhou); Validation, R.J.; investigation, M.H.; data curation, L.Z. All authors have read and agreed to the published version of the manuscript.

**Funding:** This research was funded by Key Laboratory of Environment Controlled Aquaculture (Dalian Ocean University; grant number-202302); General Project of the Educational Department of Liaoning Provincial (grant number-LJKZ0717); State Key Laboratory of Coastal and Offshore Engineering (grant number-LP2121); Scientific Research Project of the Educational Department of Liaoning Provincial (grant number-100920202019).

**Data Availability Statement:** The data presented in this study are available in article.

**Conflicts of Interest:** The authors declare no conflict of interest.

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
