# Peer review of "Experimental Study on Submerged Horizontal Perforated Plates under Irregular Wave Conditions"

_water, doi:10.3390/w15163015_

Round 1

Reviewer 1 Report

Minor revision is suggested. Please see the attachment.

Minor editing of English language is required.

Author Response

Reply to Comments of Reviewers

We thank the editor for offering us the comments. We believe that these comments have enabled us to improve our manuscript. In the following, we offer item-to-item reply to each comment and list the associated changes in the manuscript to address the comment. For clarity, the changes of revision are marked up using the “Track Changes” function in revised version.

Replies to reviewer’s comments:

Reviewer #1:

  1. 31 Please check if the relevant works can include latest references about the subject

Reply:

We gratefully appreciate for your valuable comment. In the revised manuscript, we added four references of 2021 to 2023 in lines 63 to 69 and 108 to 110.

Lines 63 to 69 of Introduction

Dong et al. (2020) investigated experimentally the forces of monochromatic and isolated waves on submerged horizontal plates in a wave tank and discussed the effect of an uneven bottom on wave loads. Huang et al. (2022) provides a novel approach that combines computational fluid dynamics (CFD) with computational solid mechanics (CSM) to dynamically simulate the fully coupled hydroelastic interaction between nonlinear ocean waves and a submerged horizontal plate breakwater (SHPB).The wave attenuation effect is found to be maximized when an SHPB has a deformation amplitude close to the incident wave amplitude. Gao et al. (2021; 2023) analyzed the mechanism of periodic submerged embankments to slow down port resonance and the impact of Bragg reflection. Stachurska et al. (2023) investigated the interaction of gravity waves with a semi-submerged rectangular cylinder of elastic bottom, and wave transmission at the structure, wave-induced pressures excreted on the plate and plate deflections were discussed.

Lines 108 to 110 of Introduction

The analyzed results of wave height, inundation depth and plate length were then used to further investigate the relationship between the wave-structure interactions. Seibt et al(2023) performed performed a full scale numerical assessment of the design of the Submerged Horizontal Plate device with the aim of improving its performance. Hayatdavoodi et al. (2023) studied the wave-induced oscillations of submerged horizontal plates, and found that the oscillation varies almost linearly with the wave height, but nonlinearly with the wave period, initial submergence depth of the plate, damping, and the spring stiffness.

References:

18.Huang, L., Li, Y., Design of the submerged horizontal plate breakwater using a fully coupled hydroelastic approach[J]. Computer‐Aided Civil and Infrastructure Engineering, 2022, 37(7): 915-932.

19.Gao, J.L. Ma, X.Z., Dong, G.H., Chen, H.Z., Liu, Q., Zang, J., 2021. Investigation on the effects of Bragg reflection on harbor oscillations. Coastal Engineering 170: 103977.

20.Gao, J.L., Shi, H.B., Zang, J., Liu, Y.Y., 2023. Mechanism analysis on the mitigation of harbor resonance by periodic undulating topography. Ocean Engineering 281: 114923.

23.Seibt F M, dos Santos E D, Isoldi L A, et al. Constructal Design on full-scale numerical model of a submerged horizontal plate-type wave energy converter[J]. Marine Systems & Ocean Technology, 2023: 1-13.

  1. 141 Character expressions like “is, CR, CT” in the line 141-142 need to be checked,especially when there is inconsistency in the display of the subscripts before and after in the text.

Reply:

We appreciate the reviewer’s important suggestion. We apologize for the careless character issues. In the revised manuscript, the mistake has been modified in the line 154, at the same time, we checked and corrected same character expression issues in the text.

  1. 128 In the 2nd paragraph of section 2.1 some spaces are missing: “0.05m, 0.063m,071m”, and the authors are requested to check whether the rest of the text are reasonably written.

Reply:

We gratefully appreciate for your valuable comment and apologize for the careless writing. In the revised manuscript, the mistake has been modified in the line 137 and check the text.

  1. 159 The authors write “[21]”, which is different from the expression in the previous Please unify the expression in the reference literature, and check the rest of the text.

Reply:

We appreciate your valuable suggestion. We apologize for the careless format issues. In the revised manuscript, we unify the expression in the reference literature according to the template in the text.

Lines 174 to 176 of Section 2.2

In the experiment, the waves superimposed by the incident wave and reflected wave are separated using the two point separation method of Goda (1976). Through comparison, it was found that the results obtained by the two point method can basically meet the accuracy requirements of this experiment.

  1. 162 “Formulas” should be centered.

Reply:

We appreciate the reviewer’s important suggestion. In the revised manuscript, the mistake has been modified in the line 178, at the same time, we checked and corrected same formatting issues according to the template in the text.

  1. 163 Do the symbol “Lmax and Lmin” in text description require italics and check the rest of the text.

Reply:

We gratefully appreciate for your valuable comment. We apologize for the careless format issues. In the revised manuscript, the mistake has been modified in the line 179, at the same time, we checked and corrected same formatting issues according to the template in the text.

  1. 165 Although Tables 2 and 3 provide the dimensions of the waves and horizontalplates for the prototype and model, it is still recommended to provide the geometric scale for the experiment in the text.

Reply:

We appreciate the reviewer’s important suggestion. In the revised manuscript, we given the geometric proportions of the experiment in the line 163.

2nd paragraph of Section 2.2

For the application of submerged horizontal plates in sea conditions, the design prototype was set at a water depth of 6.4 m, and the test water depth at 0.4 m, in accordance with the gravity similarity criterion, and the geometric scale for the experiment is 1; 16.

  1. 167 The header text format in Table 4 is not consistent with other tables. Please checkthe rest of the text and change tables uniformly according to the template.

Reply:

Thanks for your suggestion. We apologize for the careless format issues. In the revised manuscript, we checked and corrected table formatting issues in tables according to the template in the text.

  1. 182 The authors write “Significant wave height” in the line 182 and “significant wave height” in the line 445 , should this phrase be capitalized in a sentence? Please explain the reason and unify it in the text.

Reply:

We appreciate the reviewer’s important suggestion and apologize for the careless writing. “Significant wave height”should be in lowercase. In the revised manuscript, we standardized the writing in the whole text.

  1. 395 Conclusions should be curtailed summarizing only key points. Further study limitations should be added.

Reply:

Thanks for your suggestion. In the revised manuscript, we streamlined the conclusion and added restrictions on its application in the text.

6.Conclusion

In this paper, the interaction between irregular waves and submerged horizontal plates is analyzed by means of physical model tests, based on the test results of each working condition in this experiment, leading to the following main conclusions.

The impact on wave period. Solid submerged horizontal plates attenuate wave periods. The ratio of plate length and effective wavelength can affect the trend of the significant wave period. The farther away the wave is from the horizontal plate, the greater the attenuation of the wave period. The attenuation of the wave period is more significant when the submerged depth is small. Selecting an appropriate opening rate can optimize the period attenuation.

In terms of impact on wave height. The larger the submerged depth, the higher the transmission coefficient and the lower the reflection coefficient and energy dissipation, resulting in a poorer wave elimination effect. Reducing the opening rate can improve this wave elimination effect, but only to a certain extent.

In terms of impact on flow velocity. The maximum flow velocity and root mean square velocity of solid plate decreases with increasing relative plate lengths and significant wave heights. The ratio of maximum flow velocity/root mean square velocity follows a normal distribution, with a ratio of about 3.0. The relative submergence has no significant effect on the flow velocity. The presence of perforations can increase the flow velocity. However, it does not affect the maximum flow velocity and root mean square velocity of the horizontal plate significantly. The correlation between the transmission coefficient of the solid plate and the maximum velocity at the bottom of the plate is not as strong, compared to that of the perforated plate.

Further research can be conducted in the following areas in the future. For perforated horizontal plates, further discussion can be conducted on the influence of factors such as the position, shape, and size of the holes on various coefficients and energy in the future. The stress situation of the horizontal plate, especially at the connection point of the fixed horizontal plate, is also a work that can be continued in the future.

Reviewer 2 Report

GENERAL COMMENTS: The paper is very well written; it focuses on the influence of the opening ratio of the perforated horizontal plate on the wave dissipation performance. The Introduction section provides sufficient background information or context to help readers understand the significance of your research. The structure of the article is reasonable and logical. The analysis method and theory adopted in this paper are reasonable and correct. The conclusion is clear and correct and has good reference value and practical significance. However, two comments can be made to the article.

SPECIFIC COMMENTS:

1. Due to the significant impact of openings on the various characteristics of the board, it is recommended to supplement the schematic diagram of openings in the text.

2. Section 4 focuses on the influence of various influencing factors on the flow field at the bottom of the plate. Please supplement the role of this study in practical engineering.

Minor editing of English language is required.

Author Response

Reply to Comments of Reviewers

We thank the editor for offering us the comments. We believe that these comments have enabled us to improve our manuscript. In the following, we offer item-to-item reply to each comment and list the associated changes in the manuscript to address the comment. For clarity, the changes of revision are marked up using the “Track Changes” function in revised version.

Replies to reviewer’s comments:

Reviewer #2:

  1. Due to the significant impact of openings on the various characteristics of the board, it is recommended to supplement the schematic diagram of openings in the text.

Reply:

We gratefully appreciate for your valuable comment. In the revised manuscript, we added the sketch of the perforated plate in Figure 1.

Figure 1 of Section 2.1

Figure 1. Sketch of the perforated plate

  1. Section 4 focuses on the influence of various influencing factors on the flow field at the bottom of the plate. Please supplement the role of this study in practical engineering.

Reply:

We appreciate the reviewer’s important suggestion. We added the role of various influencing factors on the flow field at the bottom of the plate in practical engineering as the last paragraph of Section 4.

Last paragraph of Section 4

It can also be seen that the curves of the three opening ratios are relatively close when the relative plate length is large (greater than 0.5). This indicates that for short waves (effective wavelength less than 2 times the horizontal plate length), changing the opening rate does not have a significant effect on the wave dissipation performance.

The study of the flow field at the bottom of a horizontal plate not only provides a comprehensive understanding of the blocking effect of this structure on waves and currents, but also provides a research basis for setting up high-efficiency ocean energy conversion devices near the horizontal plate.

Reviewer 3 Report

Summary

The aim of this work is to improve the implementation of the submerged horizontal plate wave dissipation structures in (coastal) engineering. The authors studied the interaction between irregular waves and the flow field at the bottom of permeable submerged horizontal plates by means of physical model tests. More specifically, the submerged horizontal plate is used as a wave dissipation structure in engineering and the objects of investigation are its attenuation effect on the surrounding wave period and wave height. The authors found experimental results in terms of impact on the wave period, wave height, and flow velocity (the obtained results are well resumed in section 6. "Conclusion" on pages 14 and 15 of the manuscript).

the flow field at the bottom of horizontal plates.

Remarks

- At first glance, this work seems to summarize some results of a Master's thesis.

- Please, check English; a few typos were found.

- The authors claimed to have found "analytical solutions" for investigating the wave dissipation effect and the bottom flow field characteristics of permeable submerged horizontal plates through physical model trials. However, it is essentially an experimental work and the manuscript contains no analytical calculations.

- The physical interpretation of the obtained results is missing.

- The main limitations of the present approach are not discussed.

- The list of references is not exhaustive and should be completed.

- Anyhow, the work is interesting and I enjoyed reading it.

The following tips are meant to help fill in some gaps.

Suggestions

1) It is customary to specify all acronyms entering the manuscript when they appear first in the text, even when they are well known in the literature (e.g., RMS velocity = Root-Mean Square velocity).

2) Table 2. reports some parameters that, in reality, are not mentioned in the sequel of the manuscript such as Cr=reflection coefficient, S(f)=spectral function, etc. Why were they mentioned?

3) The authors mentioned that according to the requirements of the Goda et al. method, the distance between adjacent wave altimeters must satisfy requirements (1) on page 4 of the manuscript. The authors are asked to clarify these inequalities from the physical point of view. Also, show the validity of the limit values of the range appearing in the inequalities LMax< 20 \Delta l < 9Lmin.

4) Figure 2. and Figure 3. report the ratio of the effective period of transmitted waves to the effective period of incident waves vs the relative plate length parametrized by the significant wave height, the relative submerged depth, and the opening ratio. The authors mentioned that the values for the transmission period have been obtained using the average results of the three wave height meters located after the dike. Since in some cases, the amplitude of the oscillations around the mean values is quite small, for clarity, please report the error magnitude as well as the standard deviation of these measures.

5) Same issue as the previous point.

5a) Figures 4., 5. and 6. report the transmittance coefficient vs the relative plate length parametrized by Hs, dp/d, and K.

5b) Figures 7. and 8. show the positive maximum and negative maximum velocities in the X and Z directions vs the relative plate length parametrized by dp/d and K.

5c) Figures 10. and 11. show the RMS flow velocities in the X and Z directions vs the relative plate length parametrized by the opening ratio and the significant wave height, respectively.

Even for these cases, please specify the error magnitude as well as the standard deviation of these measures.

6) We come now to a key issue. We may object that the authors limited themselves to showing the experimental results obtained using their design prototype without, however, providing any physical interpretation of the results obtained. For instance, the authors state that the experimental data show that the closer the perforated plate is to the water surface, the more pronounced the attenuation of the transmittance period becomes (see Figure 2.), without however explaining why this occurs. So, to better understand the experimental findings, the authors are asked to provide also the physical interpretation of the result obtained.

7) To simulate submerged horizontal plates in (realistic) sea conditions, the authors used the prototype shown in Figure 1 where the key parameters are reported in Table 1. However, even in this case, we may object that the effectiveness of submerged wave dissipation structures depends on the site's specific characteristics, such as wave conditions, water depth, seabed composition, and coastal geomorphology. Designing structures that perform optimally across different locations can be challenging and may require extensive site-specific studies. The authors are invited to dispel this possible objection.

8) The authors did not mention the limitations of their approach. For instance,

8a) We know that submerged structures are more effective in attenuating smaller and moderate wave heights. For extremely large waves, such as tsunamis or storm surges associated with severe hurricanes, their capacity to dissipate energy may be limited.

8b) Implementing submerged structures can be technically complex and costly. Construction in the marine environment is subject to weather conditions, tidal fluctuations, and wave action, which can make the installation and maintenance of these structures more difficult and expensive. Additionally, submerged wave dissipation systems require ongoing maintenance to ensure their effectiveness and longevity. Maintenance can be challenging and expensive, especially in harsh marine environments.

The authors are invited to discuss the above points.

Conclusions

The work is interesting and topical. However, it needs some clarifications and some gaps should be filled. I invite the authors to take into account the suggestions expressed above. As regards points 7) and 8) above, it is advisable to also take into account the works that recently appeared in the literature in this regard.

Please, check English; a few typos were found.

Author Response

Reply to Comments of Reviewers

We thank the editor for offering us the comments. We believe that these comments have enabled us to improve our manuscript. In the following, we offer item-to-item reply to each comment and list the associated changes in the manuscript to address the comment. For clarity, the changes of revision are marked up using the “Track Changes” function in revised version.

Replies to reviewer’s comments:

Reviewer #3:

Reply:

  1. It is customary to specify all acronyms entering the manuscript when they appear first in the text, even when they are well known in the literature (e.g., RMS velocity = Root-Mean Square velocity).

Reply:

We gratefully appreciate for your valuable comment. In the revised manuscript, we added a specific explanation for RMS in the line 400

  1. Table 2. reports some parameters that, in reality, are not mentioned in the sequel of the manuscript such as Cr=reflection coefficient, S(f)=spectral function, etc. Why were they mentioned?2.

Reply:

We appreciate the reviewer’s important suggestion. Due to space limitations and other reasons, some of the content was deleted from the original text, but the corresponding parameters were not deleted in a timely manner. In the revised manuscript, the redundant parameters were deleted again.

Table 1 of Section 2.2

Table 1. Main experimental parameters

Main parameters

Indication Symbols

Units

Main parameters

Indication Symbols

Units

Testing water depth

d

m

Effective wavelength

λs

m

Horizontal plate submergence depth

dp

m

Wave frequency

f

Hz

Length of horizontal plate

L

m

Reflection coefficient

CR

/

Significant wave period

Ts

s

Transmittance coefficient

CT

/

Significant wave height

Hs

m

spectral function

S(f)

m2•s

Relative plate length

L/λs

/

Relative dive depth

dp/ d

/

Opening ratio

K

/

  1. The authors mentioned that according to the requirements of the Goda et al. method, the distance between adjacent wave altimeters must satisfy requirements (1) on page 4 of the manuscript. The authors are asked to clarify these inequalities from the physical point of view. Also, show the validity of the limit values of the range appearing in the inequalities LMax< 20 \Delta l < 9Lmin.

Reply:

We appreciate the reviewer’s important suggestion. Goda literature suggests that the two-point method can be used to separate reflected waves from the combination of incident and reflected waves, so formula 1 is only used to constrain the spacing of wave altimeters in front of embankments. As the later discussion in this article is limited to transmitted waves, the separation of reflected waves in this article is not emphasized.

Regarding the physical explanation of formula 1, it can be seen from Goda literature that through a series of physical model experiments, the author found that in order to separate the correct reflected wave height, the spacing between two wave height meters arranged in front of the breakwater is necessary Δ There needs to be a certain relationship between l and the incident wavelength L and frequency f. The formula recommended in the literature is used in this article.

  1. Figure 2. and Figure 3. report the ratio of the effective period of transmitted waves to the effective period of incident waves vs the relative plate length parametrized by the significant wave height, the relative submerged depth, and the opening ratio. The authors mentioned that the values for the transmission period have been obtained using the average results of the three wave height meters located after the dike. Since in some cases, the amplitude of the oscillations around the mean values is quite small, for clarity, please report the error magnitude as well as the standard deviation of these measures.

Reply:

We appreciate your valuable suggestion. The period of the transmitted wave behind the embankment is observed by three sensors behind the embankment. In the revised manuscript, the relative error amplitudes(/) of the period results of the three measuring points behind the embankment in the original Figure 2 (modified Figure 3) and Figure 3 (modified Figure 4) have been added in the last paragraph of Section 3.1 and second paragraph of Section 3.2.

Last paragraph of Section 3.1

The results of the period ratio for the 20% open-aperture plate, under the working conditions with significant wave heights, Hs = 0.1 m and Hs = 0.15 m, are presented in Figure 3. Here, the transmission period Tt is obtained using the average results of the three wave height meters located after the dike. In order to illustrate the differences in the observation results of the three wave altimeters, the relative average error amplitudes (/) of the periodic results of the three observation points behind the embankment under different working conditions were calculated. Figure 3a shows that the relative average error amplitudes under different wave conditions are between 0.42% and 7.78%, while Figure 3b shows that the relative average error amplitudes under different wave conditions are between 0.46% and 6.16%. It was found that the transmission period decreases as the relative submergence decreases. The attenuation of the transmittance period, especially for shorter waves (with larger relative plate lengths), was significant. This indicates that the closer the perforated plate is to the water surface, the more pronounced the attenuation of the transmittance period becomes.

2nd paragraph of Section 3.2

Figure 4 illustrates the variation of the transmission period of horizontal plates with different open aperture ratios for three relative dive depths with an significant wave height of 0.05 m. After calculation, the average error amplitude variation ranges for each wave condition in Figures 4a-4c are 0.53% ~ 3.56%, 0.52% ~ 14.73%, and 0.32% ~ 13.72%, respectively.

  1. Same issue as the previous point.

5a. Figures 4., 5. and 6. report the transmittance coefficient vs the relative plate length parametrized by Hs, dp/d, and K.

Reply:

We gratefully appreciate for your valuable comment. The transmitted wave height behind the embankment is observed by three sensors behind the embankment. In the revised manuscript, the relative error amplitudes(/) of the wave height results of the three measuring points behind the embankment in Figure 4 (modified Figure 5), Figure 5 (modified Figure 6), and Figure 6 (modified Figure 7) have been added in the second paragraph of Section 4.1 and first paragraph of Section 4.2 and second paragraph of Section 4.3.

2nd paragraph of Section 4.1

Figure 5 shows the variation curves of the transmission coefficient with their respective relative plate lengths for four different relative submergence conditions. The four corresponding relative dive depths are 0.14 m, 0.1 m, 0.06 m and 0.02 m, respectively. The relative average error amplitudes (/) of the wave height results at the three observation points behind the embankment were also calculated. The relative average error amplitudes in Figures 5a-5d ranged from 10.26% ~ 41.06%, 16.47% ~ 47.16%, 21.17% ~ 62.74%, and 28.04% ~ 66.67%, respectively. The relative error amplitude of this group varies greatly due to the different incident wave heights of each wave condition, which can lead to significant changes in wave heights at different positions behind the embankment. Although other groups (such as Figure 6 and Figure 7) also calculate the relative error amplitude of wave height measurement results behind the embankment, their corresponding working conditions are the same incident wave heights, so the results of the three wave heights behind the embankment have a small difference.

1st paragraph of Section 4.2

An experimental study was conducted on the horizontal plate with an opening rate K = 10% at three different wave heights and three inundation depths. The significant wave height Hs = 0.15 m is set, which is considered a relatively large value. It corresponds to five groups of periods (see Table 4. for details). Figure 6 shows the variation curves of transmission coefficient CT, reflection coefficient CR and energy dissipation with the relative plate length at three relative diving depths (submergence depth) of the horizontal plate. Since the water depth d was fixed as a constant in the tests, the corresponding inundation depths for the three relative dive depths (dp/d = 0.25, 0.15, 0.05) are 0.1 m, 0.06 m and 0.02 m, respectively. After calculation, the relative average error amplitude of each wave condition in Figure 6 is between 1.41% and 8.52%.

2nd paragraph of Section 4.3

Figure 7 presents the variation curves of the transmission coefficient, reflection coefficient and energy dissipation with relative plate length and three different open apertures, corresponding to an significant wave height of Hs = 0.05 m. After calculation, the relative average error amplitude of each wave condition in Figure 7 is between 0.61% and 12.95%.

5b. Figures 7. and 8. show the positive maximum and negative maximum velocities in the X and Z directions vs the relative plate length parametrized by dp/d and K.

5c. Figures 10. and 11. show the RMS flow velocities in the X and Z directions vs the relative plate length parametrized by the opening ratio and the significant wave height, respectively.

Even for these cases, please specify the error magnitude as well as the standard deviation of these measures.

Reply:

We appreciate the reviewer’s important suggestion. The flow velocity under the board is measured by four flow meters under the board. The flow velocity results in Figure 7 (modified Figure 8), Figure 8 (modified Figure 9), Figure 10 (modified Figure 11), and Figure 11 (modified Figure 12) are all the results of V2 measurement points. In the revised manuscript, the reasons for this selection are supplemented in Section 5.1.

Section 5.1

5.1 Measurement and uniformity analysis of plate bottom velocity field

In the experiment, the flow velocity of the four measuring points under the board (corresponding to V1, V2, V3, and V4) at each time point was measured. Based on this, the flow velocity time history curve in the time domain was plotted, and the time domain curve was transformed into the frequency domain velocity spectrum curve through Fourier transform. By comparing the velocity time history curves and velocity spectra of the four measurement points in the X-direction, it was found that the shapes of the velocity time history curves and velocity spectra of the four measurement points were basically the same, and the positions of the four spectral peaks and peaks were basically the same. This can prove that the X-direction velocity of each point at the bottom of the horizontal plate is close. In the experiment, it was found that the Z-direction velocity is generally in the order of 0.01m/s, which is relatively small compared to the X-direction velocity. Therefore, it can be said that the distribution of the flow field at the bottom of the horizontal plate is relatively uniform. One point velocity can be used as a representative for research. The V2 measurement point results will be used for analysis in the following text.

In order to study the quantitative characteristics of flow velocity, the root mean square flow velocity (calculated as formula 2) was used for discussion.

                      (2)

There, N represents a total of N flow velocity data, Vi represents the instantaneous flow velocity at each time, and Vrms represents the root mean square flow velocity.

Table 5 lists the RMS (Root Mean Square) flow rates at several representative working conditions for the V1-V4 measurement points at the bottom of the solid plate.

Table.5. RMS velocity of solid plate bottom measurement point under different working conditions

Wave height m

Period s

Submergence depth m

Root mean square velocity in X-direction m/s

Root mean square velocity in Z-direction m/s

V1

V2

V3

V4

V1

V2

V3

V4

0.05

0.85

0.14

0.012

0.011

0.015

0.018

0.005

0.004

0.004

0.005

0.05

0.99

0.1

0.022

0.024

0.018

0.023

0.010

0.009

0.011

0.010

0.1

1.23

0.06

0.072

0.065

0.078

0.063

0.029

0.022

0.025

0.024

0.15

2.67

0.02

0.112

0.102

0.126

0.118

0.032

0.028

0.036

0.034

From Table 5, it can be seen that the root mean square velocity in the X-direction and Z-direction of the V1-V4 measurement points at the bottom of the solid plate are very close under different working conditions, further confirming that the flow velocity at the bottom of the horizontal plate is basically uniformly distributed. In the following discussion, only the root mean square velocity results at the V2 measurement point will be taken for discussion.

  1. We come now to a key issue. We may object that the authors limited themselves to showing the experimental results obtained using their design prototype without, however, providing any physical interpretation of the results obtained. For instance, the authors state that the experimental data show that the closer the perforated plate is to the water surface, the more pronounced the attenuation of the transmittance period becomes (see Figure 2.), without however explaining why this occurs. So, to better understand the experimental findings, the authors are asked to provide also the physical interpretation of the result obtained.

Reply:

We gratefully appreciate for your valuable comment. In the revised manuscript, we added physical explanation of the obtained results in the last paragraph of Section 3.1.

Last paragraph of Section 3.1

It was found that the transmission period decreases as the relative submergence decreases. The attenuation of the transmittance period, especially for shorter waves (with larger relative plate lengths), was significant. This indicates that the closer the perforated plate is to the water surface, the more pronounced the attenuation of the transmittance period becomes. The reason is that the closer the horizontal plate is to the water surface, the greater the blocking effect on the upper layer wave energy, thus having a significant impact on the period.

  1. To simulate submerged horizontal plates in (realistic) sea conditions, the authors used the prototype shown in Figure 1 where the key parameters are reported in Table 1. However, even in this case, we may object that the effectiveness of submerged wave dissipation structures depends on the site's specific characteristics, such as wave conditions, water depth, seabed composition, and coastal geomorphology. Designing structures that perform optimally across different locations can be challenging and may require extensive site-specific studies. The authors are invited to dispel this possible objection.

Reply:

We appreciate the reviewer’s important suggestion. We agree with this viewpoint, so this article uses physical model experiments to understand some of the characteristics of submerged horizontal plates. In the future, numerical simulations and on-site research will be conducted to design horizontal plate structures that can operate optimally at different positions.

  1. The authors did not mention the limitations of their approach. For instance,

8a. We know that submerged structures are more effective in attenuating smaller and moderate wave heights. For extremely large waves, such as tsunamis or storm surges associated with severe hurricanes, their capacity to dissipate energy may be limited.

8b. Implementing submerged structures can be technically complex and costly. Construction in the marine environment is subject to weather conditions, tidal fluctuations, and wave action, which can make the installation and maintenance of these structures more difficult and expensive. Additionally, submerged wave dissipation systems require ongoing maintenance to ensure their effectiveness and longevity. Maintenance can be challenging and expensive, especially in harsh marine environments.

Reply:

We appreciate your valuable suggestion. In the revised manuscript, we add a paragraph between the conclusion and outlook to illustrate the limitations of the experiment.

5th paragraph of Conclusion

The maximum flow velocity and root mean square velocity of solid plate decreases with increasing relative plate lengths and significant wave heights. The relative submergence has no significant effect on the flow velocity. The presence of perforations can increase the flow velocity. However, it does not affect the maximum flow velocity and root mean square velocity of the horizontal plate significantly. The correlation between the transmission coefficient of the solid plate and the maximum velocity at the bottom of the plate is not as strong, compared to that of the perforated plate.

The conclusion of this paper only focuses on the wave conditions and corresponding structural models involved in physical model experiments. Further research is needed on the corresponding characteristics of horizontal plates under extreme wave conditions such as tsunamis and storm surges. At the same time, construction in the marine environment is affected by weather conditions, tidal fluctuations, and wave effects, and the horizontal plate is submerged below the water surface, making the installation and maintenance of the structure more difficult and expensive. In addition, the underwater wave suppression system requires continuous maintenance to ensure its effectiveness and lifespan. Maintenance can be challenging and costly, especially in harsh marine environments.

Further research can be conducted in the following areas in the future. For perforated horizontal plates, further discussion can be conducted on the influence of factors such as the position, shape, and size of the holes on various coefficients and energy in the future. The stress situation of the horizontal plate, especially at the connection point of the fixed horizontal plate, is also a work that can be continued in the future.

Conclusions

The work is interesting and topical. However, it needs some clarifications and some gaps should be filled. I invite the authors to take into account the suggestions expressed above. As regards points 7) and 8) above, it is advisable to also take into account the works that recently appeared in the literature in this regard.

Reply:

We gratefully appreciate for your valuable comment. In the revised manuscript, we added four references of 2021 to 2023 in lines 63 to 69 and 108 to 110.

Lines 63 to 69 of Introduction

Dong et al. (2020) investigated experimentally the forces of monochromatic and isolated waves on submerged horizontal plates in a wave tank and discussed the effect of an uneven bottom on wave loads. Huang et al. (2022) provides a novel approach that combines computational fluid dynamics (CFD) with computational solid mechanics (CSM) to dynamically simulate the fully coupled hydroelastic interaction between nonlinear ocean waves and a submerged horizontal plate breakwater (SHPB).The wave attenuation effect is found to be maximized when an SHPB has a deformation amplitude close to the incident wave amplitude. Gao et al. (2021; 2023) analyzed the mechanism of periodic submerged embankments to slow down port resonance and the impact of Bragg reflection. Stachurska et al. (2023) investigated the interaction of gravity waves with a semi-submerged rectangular cylinder of elastic bottom, and wave transmission at the structure, wave-induced pressures excreted on the plate and plate deflections were discussed.

Lines 108 to 110 of Introduction

The analyzed results of wave height, inundation depth and plate length were then used to further investigate the relationship between the wave-structure interactions. Seibt et al(2023) performed performed a full scale numerical assessment of the design of the Submerged Horizontal Plate device with the aim of improving its performance. Hayatdavoodi et al. (2023) studied the wave-induced oscillations of submerged horizontal plates, and found that the oscillation varies almost linearly with the wave height, but nonlinearly with the wave period, initial submergence depth of the plate, damping, and the spring stiffness.

References:

18.Huang, L., Li, Y., Design of the submerged horizontal plate breakwater using a fully coupled hydroelastic approach[J]. Computer‐Aided Civil and Infrastructure Engineering, 2022, 37(7): 915-932.

19.Gao, J.L. Ma, X.Z., Dong, G.H., Chen, H.Z., Liu, Q., Zang, J., 2021. Investigation on the effects of Bragg reflection on harbor oscillations. Coastal Engineering 170: 103977.

20.Gao, J.L., Shi, H.B., Zang, J., Liu, Y.Y., 2023. Mechanism analysis on the mitigation of harbor resonance by periodic undulating topography. Ocean Engineering 281: 114923.

23.Seibt F M, dos Santos E D, Isoldi L A, et al. Constructal Design on full-scale numerical model of a submerged horizontal plate-type wave energy converter[J]. Marine Systems & Ocean Technology, 2023: 1-13.

Round 2

Reviewer 3 Report

The authors have satisfactorily answered all the questions raised in my previous report. In my opinion, this version of the manuscript deserves to be published.